# Management Models Applied to the Human-Wolf Conflict in Agro-Forestry-Pastoral Territories of Two Italian Protected Areas and One Spanish Game Area

**DOI:** 10.3390/ani11041141

**Published:** 2021-04-16

**Authors:** Nadia Piscopo, Leonardo Gentile, Erminia Scioli, Vicente González Eguren, Ana Maria Carvajal Urueña, Tomas Yanes García, Jesús Palacios Alberti, Luigi Esposito

**Affiliations:** 1Dipartimento di Medicina Veterinaria e Produzioni Animali, Università degli Studi di Napoli Federico II, via Delpino, 1, 80137 Napoli, Italy; luigi.esposito4@unina.it; 2Parco Nazionale Abruzzo, Lazio e Molise, viale Santa Lucia, 67032 Pescasseroli (AQ), Italy; leonardo.gentile@parcoabruzzo.it; 3ASL 1 Abruzzo, via Porta Napoli, 67031 Castel di Sangro (AQ), Italy; escioli@asl1abruzzo.it; 4Departamento de Producción Animal, Campus de Vegazana, Universidad de León, 24007 León, Spain; dp2vge@unileon.es; 5Departamento de Enfermedad Animal, Campus de Vegazana, Universidad de León, 24007 León, Spain; ana.carvajal@unileon.es; 6Coto de caza Valle de Vidriales C/Jardines de Eduardo Barrón 1, Local 36, 49018 Zamora, Spain; tomasilex@hotmail.com; 7Servicio Territorial de Medio Ambiente Zamora C/Leopoldo Alas Clarín 4, 49018 Zamora, Spain; jpalaciosalberti@gmail.com

**Keywords:** wolf, conservation programs, human-wolf conflicts, wolf monitoring, protected areas, game reserve

## Abstract

**Simple Summary:**

Conservation practices in the nature of some animal species are very difficult when they are in conflict with anthropogenic activities. In order to make possible the coexistence of a predator such as the wolf with animal breeding activities in the wild, the EU has produced solid and structured legislation through the Natura 2000 network. The application of the Habitats Directive allows the various member countries to choose biodiversity management actions as long as they maintain their resilience. Our work compares two different management methods developed in Spain and Italy with the aim of evaluating a possible difference in the conservation of wolf packs present in their respective territories. The results obtained show that both in Spain and Italy, the presence of the wolf causes damage to livestock. The economic damage is quite substantial and affects, in different ways, sheep, goat, bovine, and equine breeding. Nevertheless, wolf populations are stable in Spain, where hunting is allowed, and slightly increasing in Italy, where the species is particularly protected.

**Abstract:**

Our work shows that, despite the persistence of persecutory actions, conservation activity has proved successful for the return of numerous wild mammals to different habitats, including the wolf. The human-wolf conflict is still described in all countries where the wolf is present. This is evidenced by the high number of damages on livestock, and the corpses of wolves found both in protected areas and in those where hunting is permitted. The diagnosis of road accidents, together with poisoning and poaching, are major causes of mortality. Although hunting records the highest percentage of kills in Spain, the demographic stability reported by the censuses suggests that this activity does not have a consistent influence on the Iberian wolf population’s survival. In Italy, where wolf hunting is prohibited, wolf populations are to be increasing. In some Italian situations, wolf attacks on horses seem to cause unwanted damage to foals, but they represent a very precious source of information about the habits of carnivores. A simple management plan would be sufficient to help the coexistence between the productive parts and the ecosystem services ensured by the presence of the wolf. The presence of hybrids is a negative factor.

## 1. Introduction

Before assessing the damage related to livestock and, even more so, the human-wolf conflict, it is necessary to describe the current state of the wolf presence in Europe.

Although the wolf present in Italy has been described by Altobello [1] as a typical subspecies *italicus*, and this classification is still given today [2], the discussions on its classification are not yet concluded. Issues such as wild, abandoned, and stray dogs, the genetic selection frequently made by humans to obtain new dog breeds, as well as disaster risks (e.g., fires) complicate the problem linked to the presence of wolf-dogs in natural environments [3,4,5].

The Iberian wolf (*Canis lupus signatus*) is an endemic subspecies of the Iberian Peninsula (Spain and Portugal). The scarcity of wild prey with consequent livestock predation seems to invest principally in Portugal and Spanish regions below the Duero River but does not prevent Iberian wolf survival.

### 1.1. Human-Wolf Conflict

General conflict at the European level generated by the presence of large carnivores into the agro-ecosystems has been underlined by the large number of damages that farmers-breeders complain about during the agro-food production. With reference to the recently published State of Nature 2020 report, it is essential to find workable solutions to mitigate the conflicts between human interests and large carnivores in line with EU law [6].

After the convention on biological diversity, numerous protected areas were established, which, thanks to specific management plans, have allowed the conservation of wolves and large carnivores in Europe. However, the presence of the wolf has also increased in unprotected areas, and, depending on the different management models, this predator has been responsible for greater or lesser damage to livestock farmers [7,8].

At the European level, the European Economic Community (EEC) Regulation implementing CITES (9 December 1996, n. 338) includes the Italian wolf population in Annex A, while the wolf populations of northern Spain, north of the Duero River, and Greece, north of the 39th parallel, have been placed in Annex B. This regulatory framework imposes specific authorizations for the movement of wolves, which can only be granted on the basis of careful assessments of destination site conditions.

The wolf has been protected in Italy since 3 July 1971, when hunting was prohibited by Decree of the Ministry of Agriculture. The Law 11 February 1992, n. 157 includes the wolf among the particularly protected species (art. 2, c. 1) and the President of the Italian Republic Decree (D.P.R.) 8 September 1997, n. 357, transposing the Habitats Directive (92/43/EEC), includes the wolf in Annex D, among the species of community interest that require rigorous protection.

The current regulatory framework for the protection of the wolf is reported on the new national plan for wolf conservation and management [9].

In Council Directive 92/43/EEC of 21 May 1992 about the “conservation of natural and semi-natural habitats and of wild flora and fauna,” the Spanish populations of *Canis lupus* living south of the Duero River are covered in Annex II and IV. Both indicate for these carnivores the need to conserve them through the designation of Special Conservation Areas. Conversely, the Spanish populations of *Canis lupus* which live north of the Duero River, are among the species whose taking in the wild and exploitation may be subject to management measures (Annex V).

Both Italian and Spanish regulatory framework, in line with international guidelines [10,11], gives priority to conservation at the population level. A large part of the responsibilities regarding the monitoring, management, and requalification of wildlife, the repression of crimes, the implementation of any control plans, and the reimbursement of damages are left to the regions.

### 1.2. Specific Wolf Conservation Plans in EU

The wolf is present in 20/27 EU countries (74%) with an estimated population (2012 Census) of 12,375 individuals (18 countries) and 37 identified groups in Germany/Poland (36 groups) and in Spain (1 group). The comparison with the 2016 census allows us to report a growing population trend with the exception of the Karelian, Northwestern Iberian populations which seem to have increased or stabilized. The 36 groups described in Germany have become 780–1030 individuals, while the Sierra Morena group seems to be extinct (Table 1) [10].

Comparison between two censuses allows estimating the number of individuals from 12,411 growth to a maximum of 17,199 animals. Table 2 highlights which EU countries have used special funds (LIFE+).

### 1.3. Specific Wolf Conservation Plans in Italy

The wolf population in the National Park of Abruzzo, Lazio and Molise (PNALM) would amount, according to the 1999 census data, between 30 and 38 specimens [12]. This data, compared with the latest official data produced at the national level (1100–2400 head) recognized by the European Union (census 2016), would indicate that the resident populations in the PNALM would represent 1.58–3.45% of the entire peninsular population [10].

Like other wolf conservation institutes in other states of the Union, the PNALM manages, in the town of Civitella Alfedena, a fenced structure of about 4 hectares (Area Faunistica del Lupo) within which there is a nucleus of 13 wolves.

### 1.4. Specific Wolf Conservation Plans in Spain (Castilla y Leòn)

Spanish populations of *Canis lupus* living south of the Duero River are considered protected. Conversely, the Spanish populations of *Canis lupus* present north of Duero River are managed with specific action plans of the different regions.

In the Autonomous Community of Castilla y León (CyL), all actions relating to the Iberian wolf (*Canis lupus signatus*) are coordinated by the Wolf Conservation and Management Plan implemented with the Decree 14/2016, 19 de mayo (currently judicially annulled). The plan for wolf conservation in Castilla y León is structured to keep the wolf population in a favorable and constant state. The state of conservation of the predator is possible if his presence is compatible with the traditional use of domestic livestock that together balances natural ecosystems. *Canis lupus signatus* is considered a dynamic element of rural development [13]. The wolf population in Castilla y León (94,223 km^2^) would amount, according to the 2001 census between 1000 and 1500 specimens [14], but according to the latest census (2012–2014) [15], the region would have 60% of the Iberian wolf populations present throughout the peninsula corresponding to 179 packs and 1600 wolves during the summertime with an increase of 20% [16,17]. Like other wolf conservation institutes in other states of the Union, the Castilla y León manages, in the town of Robledo de Sanabria, a fenced structure of about 23 hectares (Félix Rodrìguez de la Fuente) [18] within which there is a nucleus of 14 wolves (7 adults and 7 sub-adults).

This work aims to describe different types of wolf management in two EU member states in which there is the same conservation law. Data were analyzed in order to evaluate a possible no-conflictual presence of the wolf and breeding activities.

## 2. Materials and Methods

### 2.1. Animals

The damages reported by farmers in two Italian areas (protected areas) and in one Spanish area (game reserve) were considered, within which, as required by law, agricultural activities are allowed.

In particular, as regards PNALM, damages were described on 5064 livestock in the period 2004–2016; however, to allow comparison with the other two realities, work was carried out on 2948 heads killed by the wolf in the period 2010–2016 (Table 3).

During the 2010–2016 period, Aurunci Regional Natural Park (Aurunci), damages were reported on 892 heads (Table 3). In a specific area of the park, there is a particular situation that consists of a herd in the natural state of Esperia’s ponies. These were used to describe the dynamics of the wolf’s predatory activity. In the period 2002–2020, 2094 foals born from a number of mares varied; between 110 and 240 were studied. In the same period (2010–2016), damages were examined in the Coto de Caza Valle de Vidriales (Vidriales) and in the whole of Castilla y León (CyL) [19].

Table 1 shows the official data relating to the number of wolves surveyed in the years 2012 and 2016 and published by the environment commission of the European Union [10].

### 2.2. Study Areas

#### 2.2.1. PNALM

The study area is a national protected area in which hunting activities are prohibited but not agricultural and breeding. It is geographically localized on latitude 41°48′30.2″ N and longitude 13°47′11.3″ E and, consist of a territory of 49,680 hectares shared by three regions (Abruzzo, Lazio, and Molise) and including 25 municipalities (12 Abruzzo; 8 Lazio; 5 Molise).

PNALM extends mainly in the mountainous and pastoral territory of the Alto Sangro basin, surrounded by the Marsicani mountains to the south, while to the northeast, it is divided by the Majella, the Abruzzo plateaus, the Gizio and Tasso-Saggitario valleys. Habitat is characterized by the presence of rivers and lakes and has formed by beech woods for about 2/3 of the surface (33,286 hectares), meadows, and pastures (11,426 hectares) conditioned by the seasons that leave little room for agricultural crops (4968 hectares).

Most of the centuries-old pasture is still used today for livestock breeding. In the period of transhumance, shepherds, and their flocks, use large spaces of the protected area. In recent times, the open-air breeding of cows (meat and milk) has also developed. At the same time, it is possible to find, in the open spaces, a representative number of horses used mainly for equestrian tourism.

#### 2.2.2. Aurunci

The study area is a regional protected area in which hunting activities are prohibited but not agricultural and breeding. It is geographically localized on latitude 41°24′32.94″ N and longitude 13°29′51.66″ E and consist of a territory of 19,374 hectares included on Lazio region formed by 10 municipalities.

A specific area of 300 hectares, shared between the Municipalities of Lenola and Campodimele, has been studied for the presence of a large population of Esperia ponies which have been associated with a series of wolf attacks

Descending from the summit of Appiolo Mountain (901 m asl) toward the valley floor, you will encounter a natural landscape that develops on a carbonate relief dominated by rock outcrops, with little presence of soil and therefore of specific vegetation (chasmophytic community typical of southern Italy (*Dianthion rupicolae*, *Saxifragion australis*) and Festuco-Brometalia (xerophilous to semimesophilic-perennial polispecific grasslands dominated by hemicryptophytic grasses, generally secondary, from arid to semimesophilic)).

The agricultural areas count 1350 hectares, mostly represented by the cultivation of olives and grapevines. They are limited to the flanks and to the foothills at an altitude, normally below 400 m asl. The secular pasture is now reduced to 1162 hectares in which the last shepherds exercised, on the two sides of Lenola and Campodimele, residual breeding of native breeds endangered such as the white Monticellana goat, the Sopravvissana sheep, and the Esperia pony.

#### 2.2.3. Vidriales

The Spanish studied area is a private game reserve (Coto de Caza Valle de Vidriales) currently recognized by the Wildlife Estates agency in Brussels as a Nature Reserve, responsible not only for hunting but also for the management and conservation of wildlife in the area. It was founded in 1992 from the reorganization of land [11]. It is geographically localized on latitude 42°6′16.38″ N and longitude 5°55′56.05″ W and consists of a territory of 5493 hectares included on Castilla y León region formed by six municipalities.

The territory is divided into an agricultural area of 1350 hectares (where dry farming is the most frequently used cultivation technique, there are also large grapevine areas), and a remainder consisting of a forest area of 2840 hectares (Mediterranean woods and *Quercus ilex*); a scrub area of 1153 hectares (dominated by shrubs, small woody plants, and *Lavandula stoechas);* and an area of meadows and pastures of 150 hectares (grasses, legumes, *piperaceae*, and other herbaceous plants both natural and artificial).

The game reserve is equipped with a hunting plan that provides for the maximum killing of 380 red-legged partridge, 600 wild rabbits, 550 hares, 3 deers (2 males and 1 female), 9 roe deers (6 males and 3 females), 1 wolf and an unlimited number of wild boars in 5 hunts in a year (*monteria*).

The game reserve also provided to actions of management as: (1) corrective measures aimed at reducing limiting factors for small hunting animals; (2) choose, at regular intervals, four areas of the reserve (approximately 300–400 hectares) to be allocated to hunting activities for a period of 3–4 consecutive years. At the end of the aforementioned period, four other areas will be choosing to rotate the land subjected to hunting pressure and allow an adequate demographic restoration of the game species populations, including the wolf; (3) choose hunting type aimed primarily at hunting large animal species with particular reference to wild boar (*Sus scrofa*) and wolf (*Canis lupus signatus*); (4) improvement of habitats and protection of the entire ecosystem with particular actions of: (a) food supplement (blocks of salt, hunting products not included in the game bag, hay, straw and waste from local crops); (b) recovery of natural water sources and creation of artificial pools; (c) control of predators (huntable mammals) through hunting; (d) implement health programs aimed mainly at two species present on the reserve: rabbit (*Oryctolagus cuniculus*) and wild boar (*Sus scrofa*); (e) restocking of small game.

### 2.3. Models Applied to the Human-Wolf Conflict

In order to describe the current conflicts between anthropogenic activities and the presence of the wolf, two factors that indicate the conflict in the sampled territories were considered.

Livestock damages

When on one or more animals a predatory attack by canids occurs, inspections are organized by the competent authorities to obtain a picture of the event as complete as possible.

The operations carried out are aimed at validating the most credible hypothesis on the identification of the responsible predator for the attack based [9] on five categories of judgment to assign responsibility to the canid involved (wolf, dog, or others) and, when possible, to confirm using molecular biology tests.

Finding of dead wolves

When the presence of a dead canid is reported in the sample areas, an inspection is organized by authorized personnel who, after having filled out an identification form and taken photographs, collect the corpses and send them to the specialized centers for necropsy diagnosis and laboratory analysis.

### 2.4. Statistical Analysis

The processed data were collected in Italy (Abruzzo, Lazio and Molise regions) within protected areas where wolf hunting is prohibited and in Spain (Castilla y León region) within an area where it is possible to hunt the wolf. The comparison between two completely different realities, but both included in the Mediterranean biogeographical region, allows evaluating the human-wolf conflict, with a different but fundamental variable (hunting) on the existing wolf populations.

All the in-field collected data have been inserted in Excel tables so that they can be used with the available statistical programs (JMP^®^).

For the quantification and for the statistical analysis, the cases in which the final judgment falls into the categories “Certain canid responsibility” were considered and, of these, the cases judged in the “Probable wolf responsibility” category were then attributed to the wolf, eliminating those belonging to the category “Wolf liability excluded”.

For the statistical analysis, contingency tables were used for the comparison between the years and the attacks attributed to the wolf.

Quantitative data relating to mortality and damage to animal husbandry caused by the examined carnivores, as well as the differences between the averages of the victims, were tested with the Student’s t-test, producing significant results for the compared values (*p* < 0.01). As for the comparison of the qualitative data, the Chi-square test was used, which returned as well significant results (*p* < 0.05 and *p* < 0.01).

The Pearson correlation moment coefficient was calculated to evaluate the relationships between mortality and damage.

## 3. Results

### 3.1. General Level of Conflict in Europe

The European Parliament has recognized the importance of the role played by rural actors and the socio-economic importance of countryside activities for the conservation of biodiversity in the European biogeographic regions (ecosystem service).

The problem of large carnivores was discussed, and the related challenges and solutions to improve coexistence with anthropogenic activities. With reference to the recently published State of Nature 2020 report, it is essential to find workable solutions to mitigate the conflicts between human interests and large carnivores in line with EU law [6].

Many stakeholders are awaiting the commission’s revised guidance on strict protection to better understand how conservation and management priorities can be correctly applied toward achieving long-term coexistence with large carnivores in Europe’s densely populated and multifunctional landscapes.

The critical points to be addressed in order to identify the best practices to ensure the coexistence of large carnivores-human activities involve the answer to the following questions [20]:Which large carnivore species and populations are in need of greater conservation efforts based on the latest Member State reports?What is the appropriate scale to achieve favorable conservation status?How to assess a large carnivore’s “favorable conservation status in their natural range”?How can derogations be used in populations with unfavorable conservation status: What does recent case law say?How to deal with individual “bold” large carnivores?Favorable reference values (FRVs) and conservation status assessments at the population level: A way forward?

### 3.2. Specific Level of Human-Wolf Conflict into PNALM

Since the 1920s, the presence of wolves has been described in the areas of the National Park of Abruzzo [1]. Subsequently, between the mid-1950s and the early 1970s, it is possible to hypothesize a disappearance of the wolf in these environments. With the first conservation policies of the species [21], the census operations also began, reporting about 100 specimens throughout the Italian Peninsula [22].

Ever since in the current territory of the PNALM, the wolf has always been the object of protection, management, and conservation. Proof of this are the interventions themselves to reintroduce prey species such as deer and roe deer (carried out since the early 1970s), or the first experiences of compensation for affected farmers. Such these interventions resulted in the recovery of the wolf not only under a demographic perspective but, above all, as to ecological aspects in the territory of the park and in the neighboring Apennine areas [23].

During the 1980s and 1990s, the wolf was the subject of various monitoring programs. Since 2005 a new research project involved on large carnivores in the park aims at defining the numerical entity, the genetic identity, the social and territorial organization of the populations, and the food ecology through the analysis of predations operated on wild and domestic species [24].

Analysis of the samples studied (PNALM)

The open conflict between the wolf and the breeders is confirmed by the analysis of the causes of death of the predator reported within the National Park from 1999–2017 (Table 4). In order to evaluate the differences between the three realities considered, the data relating to the mortality of wolves and the predations carried out by them in the period 1999–2017 are taken. However, where data are available outside this period, it was decided to describe them to corroborate what was reported in a different period.

In Table 4, during the period 1999–2017, the causes of mortality of wolves found dead (corpses or remains, *n* = 139) in the PNALM are reported [25,26].

Figure 1 reports the trend of the percentages of dead wolves and damage recorded in the PNALM, comparing them among years.

In Figure 1, it is also possible to see a negative correlation: as the damage caused by the wolf decreases, the mortality rate increases (r = −0.0567).

The analysis of Figure 1 allows us to consider relatively uniform the mortality of wolves and the damages percentage within the Italian National Park. There are no statistical differences between the years, with the exception of the years 2007, 2010, 2012, and 2013 (*p* < 0.01), when the number of dead wolves is higher than 10, up to a maximum of 28 in 2013.

The damages are repeated with the same frequency, and significant differences (*p* < 0.01) are reported only for the years 2008, 2009, 2011, 2015, and 2016 compared to the other years considered.

Finally, Figure 2 shows the correlation (R^2^ = 0.506) between the percentage of dead wolves and the reported damage.

The major cause of mortality of wolves (males 48%; females 38%; indeterminate 14%) is represented by injuries (accidents and traumas 47%).

Among the injuries and trauma, the highest percentage is attributed to road accidents (53%), while 24% is caused by gunshot wounds or traps and 23% by aggressions between congeners or from other canids [25].

Apart from the undetermined causes of death, which represent the highest share (31%), poisoning from organo-chlorine (21%) and organophosphates (18%) are the most represented features in percentage. Following, in descending order there are: chemical compounds (9%); alkaloids, and carbamates (6%); coumarins, thiophosphoric ester, and arsenic (3%).

Among the pathogens identified at necropsies and laboratory analysis, the presence of viruses and endo and ectoparasites are described. To these etiological agents is not always possible to ascribe the cause of death.

The analysis of the overall situation of the wolf presence in the PNALM allows to estimate certain stability of the National Park population and confirm what is reported by the EU and International Union for Conservation of Nature (IUCN) data which consider the *Canis lupus* population in a status of least concern (LC), with a tendency to stability [27].

In the three study areas, in order to homogeneously compare the dynamics and wolf predatory preferences on domestic livestock, the data were sorted for the same period, 2010–2016. The complaints received by breeders amounted to 2948 head of livestock. The attacks were certainly attributed to the wolf for 62%. The type of predation is slightly different between the years considered.

The prey of choice appears to be the sheep, followed by the goat, the bovine, and the horse (Table 5).

### 3.3. Specific Level of Human-Wolf Conflict into Aurunci

As already described for the PNALM, in the 1920s, the wolf was present in all the Italian mountains [28,29], so it can be said that it was also present on the extreme offshoot of the Lazio Antiapennines of the Ausoni Mountains and the Aurunci Mountains.

By the same principle, it is possible to hypothesize a disappearance of the wolf in these environments between the mid-1950s and the early 1970s [22].

Surveys carried out in 2000–2001 exclude the presence of stable wolf nuclei into the Aurunci. However, the research carried out between 2004 and 2005 [30,31] indicated the presence in the entire range of the Regional Natural Park of the Aurunci Mountains of 3–4 individuals of the *Canis lupus* species.

These reports seem to be the first after those of 1985 and refer to a spontaneous return of the canid without having resorted to reintroduction. In the same year, 2005, the Regional Natural Park of the Aurunci Mountains Authority began an indemnity campaign for damage caused by predation to breeders in the protected area.

From that moment on, the reports of predation and sighting of wolves have not stopped throughout the park area, with particular reference to Monte Petrella (1500 m asl) [32] and Monte Appiolo (901 m asl) [33].

Analysis of the samples studied (Aurunci)

Also, for Aurunci Park, we have analyzed the data relating to the mortality of wolves and the predations carried out by them in the period 1999–2017 and evaluated the differences between the other two realities. However, as in the case of PNALM, it seemed useful to describe a particular human-wolf conflict that we have been able to study for a long period of time. During the period 1999–2017 (Table 4), five dead wolves were found in the Regional Natural Park of the Aurunci Mountains (one for accident, 2005; two for unknown causes, 2006, 2007). In the same period, in-field checks and expert investigations indicated the simultaneous presence of stray dogs. Stray dogs are one of the main issues involved in the conservation of endangered and protected species such as wolves. Therefore, in 2008, the Park Authority launched the census of stray dogs [34]. In the years 2014 and 2016, two wolves dead for poaching were found.

The reasons that lead to the realization of the specific work, as below reported, have their roots in the framework law on protected areas 394/1991. The human presence in the agro-forestry-pastoral habitat guarantees therefore, a timely application of One Health-related environment management dynamics.

The Sustainable Development Agenda [35] reiterates the need for restoration methods suitable for achieving integration between man and the natural environment, including by safeguarding anthropological, archaeological, historical, and architectural values and traditional activities.

In the area of the park, there is a consistent representation of an endangered equine breed: Esperia’s pony. These are animals whose original breeding area includes the Aurunci Mountains and the Ausoni Mountains and in which the result of a very rigid natural selection is evident, which has shaped their contained forms (height at the withers 132–138 cm) and the extreme rusticity. The survival of this breed, able to use fodder resources that are difficult to reach and, in any case otherwise lost, appears as the only condition to be able to maintain the human presence in territories that seem increasingly destined for abandonment and degradation [36].

In the study area, data relating to predatory attacks by wolves on domestic livestock in the period 2010–2016 were collected. The complaints received by breeders amounted to 892 head of livestock. The attacks were certainly attributed to the wolf for 70%. The type of predation is slightly different between the years considered. The prey of choice appears to be the sheep, followed by the goat, the horse, and the bovine (Table 5).

The case of the reproductive tendency of the Esperia’s ponies explains the trend over time of the predatory attitude of the wolf (Aurunci)

The sample on which we worked consists of a group of 150 Esperia’s pony (15% of the Italian population estimated at about 1000 mares) reared in the wild on an area of 300 hectares of Monte Appiolo. The mares’ pregnancies were followed from 2002 to 2020.

The births take place outdoors and, normally, without human help, between February and October of each year. During the same period, neonatal and foal mortality rates at different ages (within the year) were noted. Attacks by canids on foals were analyzed, reported by the shepherd.

Table 6 shows the number of births registered from 2002 to 2020 and the relative mortality of Esperia’s foals reported by the breeder. The results indicate that from the year 2002 to 2006, the number of mares increased, stabilizing on 150 breeding mares until the year 2016. A new increase from 200 to 240 mares is recorded in the period 2017–2020.

Obviously, during the same period, the number of births increased, which, in the entire period considered, was 110.21 ± 8.31 per year. The neonatal mortality of foals ranges from zero to 15 in the period 2002–2009 (7.63 ± 4.95) and 7.16 ± 2.34 in the period 2010–2020, where mortality of foals was also ascribed to predation by canids.

Predation begins to be reported in 2010 and becomes significant in 2011 (5.79 vs. 19.20% *p* < 0.01). The increase in mortality is significantly high in 2012 (19.20% vs. 34.86%; *p* < 0.05) and 2013 (19.20% vs. 53.57%; *p* < 0.01).

This last year is the first in which was described the highest number of foals dead by an attack of wolves. The mortality rate will decrease compared to 2013 but will remain constant (24.29 ± 8.08%) and, in any case, significantly high compared to 2010. In the last four years, predatory attacks have again increased the mortality rates (39.55% ± 11.16%), and they peak (54.64%) comparable to that recorded in 2013 (Table 6).

Foals birth alive (vivinatality) was highest in 2002 (97.27%), maintained an average of 82.42% between 2003 and 2011 and, then collapsed between 2012 and 2016 to 73.07%. In the last four years, vivinatality was the lowest in the entire study period (43.33%).

The analysis of the above data, combined with the information provided by Table 5, allows us to hypothesize a significant influence of predations on the reproductive results of the Esperia’s ponies.

An important observation (Figure 3) about predatory attacks consists of a significant seasonal pattern. All attacks on horses from 2002 to 2020 in the various months in which they occurred were classified. The attacks begin, in fact, in the month of April, during which the attention of predators would be directed toward foals born in February (that is, about two months old). The monthly mortality continues with an ascending curve until the month of June and then gradually decreases until it reaches zero in the month of December. The total percentage of foals’ deaths caused by predation, compared to the number of births, is 24.93%. The mortality rates begin to increase from April (11.30%) to become more consistent and descendant in May (31.03%), June (28.16%), July (13.60%), and August (11.69%). Percentages that are reduced in residuals in September (2.30%), October (0.96%), and November (0.96%).

The analysis of the data also allows us to observe that in the months of December, January, February, and March, there are no predatory attacks, which would indicate a different trophic choice by the predators, probably justified by the absence of foals older than two months but less than six months of age, in the predation area.

Evaluating the sum of the number of dead foals, divided into the different months of the year, it is clear that, while the births are concentrated between February and October, predations are concentrated between April and November.

The advanced considerations would indicate a seasonal presence of wolves or groups of wolves in the area under study, which would begin to arrive in March–April and leave at the end of November.

Unfortunately, due to the lack of the authorizations, it was not possible, in this work, to indicate how many wolves or how many groups have occurred in the area and whether there have been litters, all elements that would explain the dynamics of the recorded predations and would make it possible to identify fundamental management actions to preserve a breed of a horse in danger of extinction and in the same way a wild species, the wolf, that does not deserve human inattention, especially in an area under protection.

### 3.4. Specific Level of Human-Wolf Conflict into Vidriales

Table 7 shows the causes of mortality of wolves found dead between 1999 and 2004 in the Castilla y León [18] and into Vidriales between 1999–2017.

The biggest difference that is immediately noticed is the high percentage of wolves killed for hunting activities (42.73%), which in the Spanish region is allowed in derogation from the Habitats Directive. In addition, the samplings that are carried out by the park rangers for population control operations (3.14%).

Conversely, like the two Italian study areas, the other causes of death are represented by road accidents (7.98%), unknown (2.14%), and poisoning (0.85%) causes.

Although hunting activities are allowed in Vidriales, in the reference period, no wolf hunts were organized. However, in the study area, the presence of the wolf is evidenced by the discovery of one dead subject for unknown causes in 2015.

The management of the hunting reserve is based on European legislation (Directive 92/43/EEC) and on the law of Castilla y León, number 4 of 12 July 1996, which exercises exclusive jurisdiction as an autonomous region.

Title II, in Article 7 defined as game species those listed in Annex I. Among these, the wolf (*Canis lupus*) is mentioned as the big game species north of the Duero River.

Annex II indicates that wolf hunting can be carried out from the fourth Sunday in September until the fourth Sunday in February of the following year.

The owner of the agricultural land that can be used for hunting purposes (Title IV) can request the consideration of €2500–3000 for a wolf killed.

However, the use of this practice would not negatively affect the size of the populations residing in the region. In fact, Castilla y León harbors around 60% of the total Iberian wolves (14). At least 179 packs were found in the last census 2012–2014 [16,17]. Due to its close presence in rural environments, persecutory acts against the wolf have also increased in Spain. In fact, in large areas of the Iberian territory, it is common to find dead animals from poaching or poisoning [19].

The reports received from Vidriales count no attack by wolves on livestock.

This result is due to two orders of factors:(a)the type of farming.(b)the availability of natural prey.

In Vidriales, there is the intensive breeding of the dairy sheep organized in modern stables that house from 250 to 500 sheep. The animals go out to pasture for about 4 h in the morning, strictly controlled by the shepherd accompanied by at least 3 Spanish mastiff dogs and 2–3 shepherd dogs (perro carea) to return indoors for milking and the rest of the day. Other livestock (goats, cows, and horses) are scarce.

In all environments that make up the coto de caza, there are wild boars and roe deers, which represent the major trophic resource of the wolves present in Vidriales.

## 4. Discussion

The human-wolf conflict was measured by estimating the number of domestic livestock preyed and the number of wolves found dead in three experimental areas. The effects of predations have significantly affected the mortality of some animal species. As a consequence, it is possible to hypothesize a significant influence of predations on the reproductive results, for example, of the Esperia’s Pony.

The greater number of available and analyzed data relating to PNALM allows us to hypothesize a close relationship between the number of predated animals and the number of wolves found dead.

Conversely, in the other two realities, in contrast to predations, there are no significant reports of dead wolves.

The analysis of the overall data of the entire region of Castilla y León allows us to describe the causes of mortality of the recovered wolves. The results show that these causes of mortality are comparable in the two EU countries.

The experience gained in about 20 years in the two Italian parks indicates that the promotion of scientific studies in protected areas, through results, is able to explain in a consistent manner the behavior of predators and their presence compatible in agro-forest-pastoral territories. The acquisition of this knowledge is able to provide the management entities with the appropriate tools for the coexistence between sustainable anthropic activities and the protection of the wolf. All of this in a healthy and balanced environment, as recommended by the United Nations in the context of Agenda 2030, in accordance with the directives inherent to the One (Digital) Health paradigm [37].

## 5. Conclusions

The increase in damage which, although unable to prove it, could be ascribed to a demographic increase of the wolf, has caused two emerging problems linked to each other: (1) the onset of new conflicts with humans (especially in territories, as in Italy, where the absence of the carnivore for a long time has meant that the memory of living with the predator was lost); (2) the doubt of the genetic purity of domestic fauna predators.

Both problems exacerbate the anthropic conflict and contribute to what was seen in the second half of the last century or to the progressive abandonment of the territories with the consequent deterioration of the trophic qualities available.

Our work highlights that the actions to protect farms, the knowledge and containment of the persecution, as well as the actions aimed at individual species, included action of well-being assessment [38], have been able to bring back many species of mammals, as also demonstrated by European Rewilding Network [7]. It also demonstrates that veterinary assistance and wildlife technicians represent one element that, along with the support of operators in the agro-forestry-pastoral sector and the multidisciplinary work, favors the increase of already existing packs and allows the correct settlement of new populations. It is critical (at both national and international level) the presence of a good collaboration among the scientific world, the animal breeding, and the administration entities.

## Figures and Tables

**Figure 1 animals-11-01141-f001:**
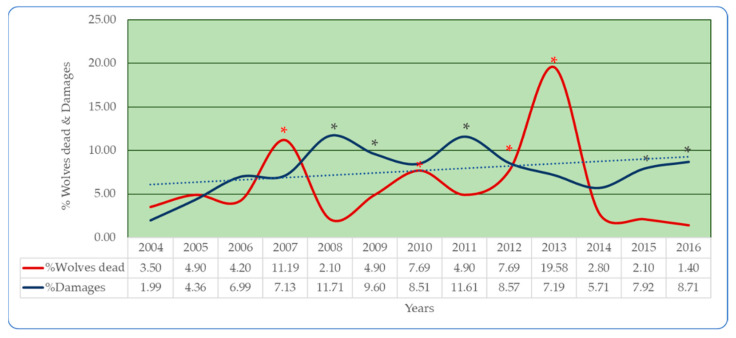
Relationship between percentages of dead wolves and damage caused by the canid. The asterisk indicates significant differences (*p* < 0.01) between the years compared.

**Figure 2 animals-11-01141-f002:**
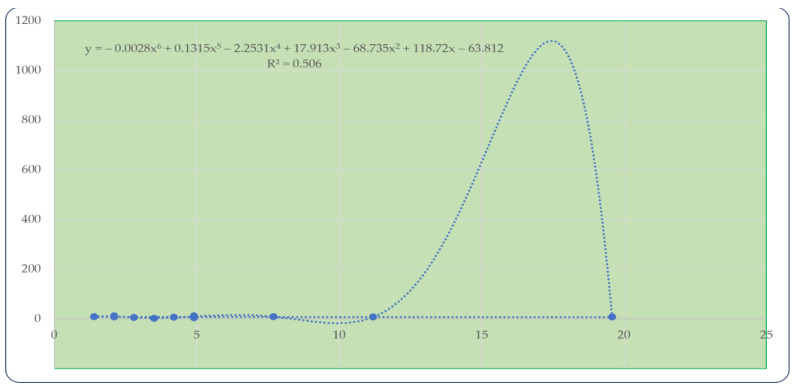
Correlation between dead wolves and damage caused by the canid.

**Figure 3 animals-11-01141-f003:**
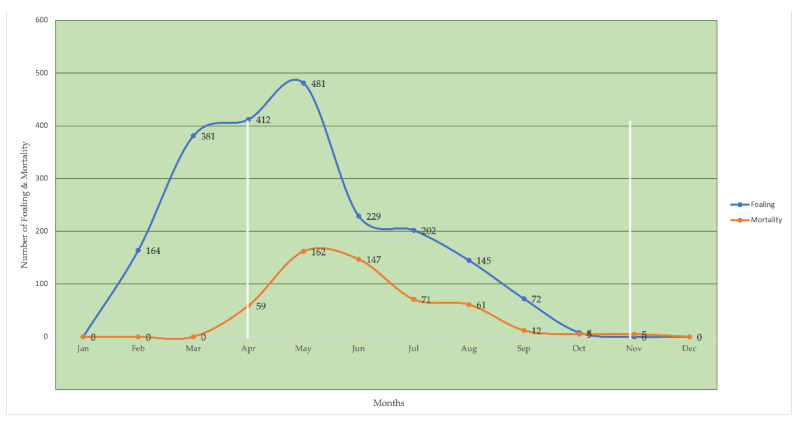
Foaling (vivinatality) and monthly mortality in Esperia’s ponies among the months.

**Table 1 animals-11-01141-t001:** Distribution and presence of the wolf in Europe and in the EU member states.

Population Name	Countries	Size		Trend
		Census 2012	Census 2016	
Scandinavian	Norway, Sweden	260–330	c. 430	Increase
Karelian	Finland	150–165	c. 200	Stable to increase
Baltic	Estonia, Latvia, Lithuania, Poland	870–1400	1700–2240	Stable
Central European lowlands	Germany, Poland	36 groups	780–1030	Increase
Carpathian	Slovakia, Czech Republic, Poland, Romania, Hungary, Serbia	3000	3460–3849	Stable
Dinaric-Balkan	Slovenia, Croatia, Bosnia and Herzegovina, Montenegro, the former Yugoslav Republic of Macedonia, Albania, Serbia (incl. Kosovo), Greece, Bulgaria	3900	c. 4000	Unknown
Alps	Italy, France, Switzerland, Austria, Slovenia	280	420–550	Increase
Italian peninsula	Italy	600–800	1100–2400	Slightly increasing
		Census 2007		
NW Iberian	Spain, Portugal	2500	2500	Unknown
Sierra Morena	Spain	1 group	0	Extinct

Estimated number of wolves in Europe ~12,375 (2012)–14,590–17,199 (2016) in bold EU countries. http://ec.europa.eu/environment/nature/conservation/species/carnivores/conservation_status.htm (accessed on 22 February 2021).

**Table 2 animals-11-01141-t002:** Specific conservation plans (LIFE+) for the large carnivores in EU countries (up to 2018).

	Country	*n*.		Country	*n*.
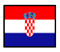	Croatia	1	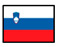	Slovenia	3
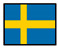	Finland	1	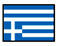	Greece	4
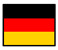	Germany	1		Romania	5(1)
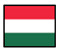	Hungary	1	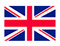	U.K.	
	France	2	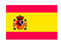	Spain	5
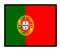	Portugal	3		Italy	21

**Table 3 animals-11-01141-t003:** Number of livestock heads attacked by wolves in the three study areas (National Park Abruzzo, Lazio, and Molise = PNALM; Regional Park of Aurunci Mountain = Aurunci; Coto de Caza Valle de Vidriales = Vidriales).

**PNALM**	**2010**	**2011**	**2012**	**2013**	**2014**	**2015**	**2016**
Sheep	172	235	174	146	115	160	176
Horse	48	65	48	40	32	44	49
Goat	129	176	130	109	87	121	132
Cow	82	112	82	69	55	76	84
Total	431	588	434	364	289	401	441
**Aurunci**	**2010**	**2011**	**2012**	**2013**	**2014**	**2015**	**2016**
Sheep	45	54	47	43	38	45	49
Horse	36	48	40	47	32	27	31
Goat	30	49	43	34	33	41	42
Cow	0	3	5	7	6	8	9
Total	111	154	135	131	109	121	131
**Vidriales**	**2010**	**2011**	**2012**	**2013**	**2014**	**2015**	**2016**
Sheep	0	0	0	0	0	0	0
Horse	0	0	0	0	0	0	0
Goat	0	0	0	0	0	0	0
Cow	0	0	0	0	0	0	0
Total	0	0	0	0	0	0	0

**Table 4 animals-11-01141-t004:** Number of wolves founded dead in the samples studied areas of Italy (National Park Abruzzo, Lazio, and Molise = PNALM; Regional Park of Aurunci Mountain = Aurunci).

Period 1999–2017	PNALM	Aurunci	Total
Accidents	50 (36%)	1 (20%)	51
Hunting	0	0	0
Control	0	0	0
Unknown	43 (31%)	2 (40%)	45
Poisoning	31 (22%)	0	31
Poaching	15 (11%)	2 (40%)	17
Total dead wolves	139	5	144
Damages to livestock	2948	892	3840
(in heads dead 2010–2016)			

**Table 5 animals-11-01141-t005:** Percentage of domestic animals preyed by wolves in Italian studied areas. (National Park Abruzzo, Lazio, and Molise = PNALM; Regional Park of Aurunci Mountain = Aurunci).

Year	Protected Area	Sheep	Goat	Cow	Horse
2010	PNALM	39.91	29.93	19.03	11.14
	Aurunci	40.38	26.92	32.69	0.00
2011	PNALM	39.97	29.93	19.05	11.05
	Aurunci	35.34	31.90	31.03	1.72
2012	PNALM	40.09	29.95	18.89	11.06
	Aurunci	34.42	31.82	29.87	3.90
2013	PNALM	40.11	29.95	18.96	10.99
	Aurunci	33.17	25.63	35.68	5.53
2014	PNALM	39.79	30.10	19.03	11.07
	Aurunci	34.84	30.32	29.03	5.81
2015	PNALM	39.85	30.08	19.05	11.03
	Aurunci	37.50	33.93	22.32	6.25
2016	PNALM	39.95	30.02	18.96	11.06
	Aurunci	36.50	31.73	24.04	6.73

**Table 6 animals-11-01141-t006:** Vivinatality of newborn foals; mortality numbers and percentage of Esperia’s ponies due to canid attacks in the period 2002–2020.

		Vivinatality		Mortality					
Year	Mares	Birth		Total		Neonatal		Preyed	
	*n*	*n*	%	*n*	%	*n*	%	*n*	%
^ 2020	240	97	40.42	62	63.92 ***	9	9.28	53	54.64
2019	240	103	42.92	48	46.60 **	10	9.71	38	36.89
2018	240	108	45.00	48	44.44 **	6	5.56	42	38.89
2017	200	90	45.00	30	33.33 **	5	5.56	25	27.78
2016	150	106	70.67 *	25	23.58 *	4	3.77	21	19.81
2015	150	108	72.00 *	25	23.15 *	4	3.70	21	19.44
2014	150	113	75.33 *	45	39.82 **	7	6.19	38	33.63
^ 2013	150	112	74.67 *	71	63.39 ***	11	9.82	60	53.57
2012	150	109	72.67 *	46	42.20 **	8	7.34	38	34.86
2011	150	125	83.33 *	36	28.80 **	12	9.60	24	19.20
§ 2010	150	121	80.67 *	17	14.05 *	10	8.26	7	5.79
2009	150	110	73.33 *	12	10.91 *	12	10.91	0	0.00
2008	150	115	76.67 *	15	13.04 *	15	13.04	0	0.00
2007	150	120	80.00 *	14	11.67 *	14	11.67	0	0.00
2006	150	120	80.00 *	12	10.00 *	12	10.00	0	0.00
2005	130	112	86.15 **	10	8.93	10	8.93	0	0.00
2004	120	108	90.00 ***	6	5.56	6	5.56	0	0.00
2003	120	110	91.67 ***	1	0.91	1	0.91	0	0.00
2002	110	107	97.27 ***	0	0.00	0	0.00	0	0.00

Asterisks indicate significant differences among the years (* *p* < 0.05; ** *p* < 0.01; *** *p* < 0.001); the differences among the percentages were calculated using the chi-square test; § year in which the breeder begins to report canids attacks; ^ years in which the highest mortality is recorded since the beginning of canid attacks.

**Table 7 animals-11-01141-t007:** Number of wolves found dead in the areas of Spain.

Period 1999–2017	Castilla y León *	Vidriales
Accidents	56 (7.98%)	0
Hunting—Legal control	300 (42.73%)	0
Unknown	15 (2.14%)	1
Poisoning	6 (0.85%)	0
Poaching—Illegal control	325 (46.30%)	0
Total dead wolves	702	1
Damages to livestock (Attacks paid for dead livestock 2010–2016)	6813	0

* from the year 1999 to 2004.

## Data Availability

Publicly available datasets were analyzed in this study. This data can be found here: https://wolf.fandom.com/wiki/Wolf-dog_hybrid (accessed on 22 February 2021); https://medioambiente.jcyl.es/web/es/medio-natural/plan-conservacion-gestion-lobo.html (accessed on 22 February 2021); https://www.minambiente.it/comunicati/lupo-il-nuovo-piano-di-conservazione-e-gestione-prevede-la-prevenzione-attiva-e (accessed on 22 February 2021); https://ec.europa.eu/easme/sites/easme-site/files/italy_update_it_june20.pdf (accessed on 22 February 2021); https://ec.europa.eu/environment/nature/conservation/species/carnivores/conservation_status.htm (accessed on 22 February 2021); https://www.minambiente.it/pagina/elenco-ufficiale-delle-aree-naturali-protette-0 (accessed on 22 February 2021). In-field data presented in this study are available on request from the corresponding author.

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
