# Peer review of "Management Models Applied to the Human-Wolf Conflict in Agro-Forestry-Pastoral Territories of Two Italian Protected Areas and One Spanish Game Area"

_animals, 2021, doi:10.3390/ani11041141_

Round 1
Reviewer 1 Report
I enjoyed reading this paper and there is quite some potential to turn this into an important and relevant piece of work with clear conservation, policy and management implications. But in my view it does require quite a bit of work and more attention to detail.
The abstract and simple summary are clear and give a good overview of the content of the paper.
The introduction needs to be rewritten – this is now a taxonomic overview of the wolf and while perhaps correct, it is largely irrelevant for the purpose of this study. Give an overview of the main points you try to address in this study – that is management models, human-wildlife confect, how wolf can persist in agro-forest areas, how protected areas have a role to play in wolf or large carnivore conservation in Europe.
End the Introduction with some very clear aims or hypotheses. You can frame this along the subheadings you now present in the Results
The methods are good when it comes to the animals, the study areas (although this should be written in a more coherent fashion and more consistent between the three areas – now it seems as if the entry for Vidriales is written by a different person – make them more the same), and the analysis is alright. However, it does not align well with the Results – there are section in the Results where it is not clear how you obtained it and this needs to be added in the Methods.
The Results are interesting but 3.1 does not naturally follow from the Introduction (so make a review of the protection by regulatory framework one of your aims).
3.2 is interesting but again make it clear that this is an aim of the study
Discussion
A lot of information from the Results can be and should be included in the Results. There is now a clear imbalance between these sections. The discussion should make reference to the literature – now there is none. This requires a rewrite.
Conclusion – some of this could go into the Discussion and then keep this section focussed on your findings and your conclusion
References are a bit messy (for instance reference 1 is not a refernece; lots if references have missing information, are incomplete, etc). There is also an overreliance on Italian literature – there is lot of data / papers / reports / books out there written in English that are highly relevant for the present study and that allow you to see the bigger picture.
Figures: there all need a lot of work – better legends, better explanation of the axis, make them appear more uniform. You can probably get rid of half the figures and retain all the necessary information.
Figure 4 is really not needed – all this can be reported more effectively in the text.
Figure 7 not clear what the y-axis represents
Tables: this comes across as really messy – make them appear all the same; have one person do it, now it seems as if five people all have made different tables (and figures) and these are here combined. Perhaps model them all after Table 8. This one is clear.
See figures above on how to improve them. You can probably get rid of half the tables and retain all the necessary information.
Author Response
The Authors thank the referees for their comments, suggestions and reviews.
"Please see the attachment"

Reviewer 2 Report
Congratulations, a good job, very interesting to read, but in general it presents some errors that could greatly improve the results of the research and would help to disseminate the results and be able to be used:
General comments: In general, they should improve and unify the presentation of the results. And from the tables, below, some specific comments but in a general way, statistical analysis and information would be lacking. They should all be self-explanatory. I think you should review them all carefully.
Generally speaking, the structure is complicated to follow. Example (3.2.3.1)
Authors "y" appears before Tomas, please check the order and if so, change the y. I would try to synthesize and rearrange the tables so that they provide more information in less space. For example 11, Figure 9 Figure 1, they could be a single figure making even a scientific contrast between them.
Many data of the results are duplicated, written and in tables / figures, please try to summarize it, it makes reading much easier.
Figure 1: Does it refer only to Life-type projects? To funded projects? I think it should be specified.
Figure 2: This more than a figure is really a table, and although it is not properly it should have the format of all tables. With a reference.
Table 4: Here I do not understand well, I do not understand why there are different numbers.
Table 5: What does PNALM mean? Tables should be self-explanatory.
General comment tables: Sometimes the titles are in italics, sometimes in bold, sometimes in both, please, unify criteria.
Figure 3: What do the letters stand for? If it is a comparative statistic, it should appear reflected, along with the standard error. Also appears "%" refers to the percentage of the total census? Tables should be self-explanatory.
Figure 4: Have they made a statistical comparison? Perhaps it would be important to scientifically endorse which is the greatest damage caused.
Figure 6: Same comment as figure 3.
Figure 7: Same comments.
Figure 8: Sometimes they put the title at the top and sometimes they don't.
Figure 8: is there some kind of statistical analysis?
Table 6 and Table 7: What do those colors mean? I repeat that they must be self-explanatory.
Figure 10: How has this trend been done?
Table 8: Dagame is not capitalized.
Author Response
The Authors thank the referees for their comments, suggestions and reviews.
"Please see the attachment."

Round 2
Reviewer 1 Report
I had a look at the revised version. I still think the English needs improvement with many sentences being awkwardly constructed and difficult to interpret.
For me the Introduction has not improved much - there is lots of information on the wolf, but we need background information on the wider issues the authors try to address. The newly added paragraphs do improve the Introduction somewhat, but overall this section feels disconnected from the remainder of the paper. The same is true for the majority of the remainder of the paper (Methods, Results, Discussion) where I see very little improvement.
The authors indicate that they have worked on the tables but some of the figures (1 and 2) are in essence tables and they have different fonts, different font sizes, contain non-English words (grupo, grupos, Pais), "the former Yugoslav Republic of Macedonia" is now knowns as North Macedonia, in figure 2 the UK conservation plan is one that deals with Romania not with the UK.
The figures, including the new ones, show absolutely no coherence in terms of style, font, layout, content, and all in all it comes across as very messy.
There are quite a few single sentence paragraphs - this works for a newspaper but not for a scientific article.
I appreciate the work that has gone into preparing this manuscript but in my opinion it still needs a lot of work to make it publishable.
Author Response
We thank the reviewer for his comments, correction and suggestions which are reported below:
- I had a look at the revised version. I still think the English needs improvement with many sentences being awkwardly constructed and difficult to interpret.
- The article was rewritten following the suggestions of the referees and the English version of the final paper was checked again.
- For me the Introduction has not improved much - there is lots of information on the wolf, but we need background information on the wider issues the authors try to address. The newly added paragraphs do improve the Introduction somewhat, but overall this section feels disconnected from the remainder of the paper. The same is true for the majority of the remainder of the paper (Methods, Results, Discussion) where I see very little improvement
- The authors, after having accepted the referee's suggestions at the first review, have modified the introduction in a small part since they are convinced that:
1) Basic information about the current official description of wolf types and subtypes helps in choosing the best management to limit the wolf-breeders conflict.
2) The referee's observation at the first review, about the poor description of the aims of the work was correct and, therefore, authors inserted a new paragraph. However, the small size of the introduction is a choice of the authors who preferred to detail the following chapters and paragraphs.
With the aim of improving it, all the work was once again modified in its different parts that compose it (Materials and Methods, Results, Tables, Figures, Discussion).
- The authors indicate that they have worked on the tables but some of the figures (1 and 2) are in essence tables and they have different fonts, different font sizes, contain non-English words (grupo, grupos, Pais), "the former Yugoslav Republic of Macedonia" is now knowns as North Macedonia, in figure 2 the UK conservation plan is one that deals with Romania not with the UK.
- The authors transformed Figures 1 and 2 into Tables 2, 3, 4 by unifying the characters and sizes and eliminating the non-English words. We apologize for this.
We apologize but we cannot change the wording shown in former figure 1 now Table 2 "the former Yugoslav Republic of Macedonia" in North Macedonia as this wording is officially reported on the site https://ec.europa.eu/environment/ nature / conservation / species / carnivores / conservation_status.htm
Figure 2 has become Tables 3 and 4. The authors report that the LIFE + to which the referee refers was funded to the UK for the benefit of Romania. Therefore, in Table 4 it was decided to insert the UK conservation plan in correspondence with the Romania box.
- The figures, including the new ones, show absolutely no coherence in terms of style, font, layout, content, and all in all it comes across as very messy.
- The authors apologize and have reorganized the figures in terms of style, character and layout.
- There are quite a few single sentence paragraphs - this works for a newspaper but not for a scientific article.
- The subsections have been reorganized
- I appreciate the work that has gone into preparing this manuscript but in my opinion it still needs a lot of work to make it publishable.
- The authors are grateful for appreciating the effort made to demonstrate that in the conservation of the wolf but of biodiversity as a whole, common rules are not enough, but it is necessary to verify in reality what happens.
In this review, the Italian and Spanish researchers made efforts to improve the work and make it more understandable. We hope we have succeeded.
The authors thank the Referees for valuable suggestions and review.
Best regards
Nadia Piscopo

Reviewer 2 Report
Congratulations. The authors have made the changes I suggested and from my point of view, the work presented has clearly improved.
Author Response
- Congratulations. The authors have made the changes I suggested and from my point of view, the work presented has clearly improved.
- The authors thank the reviewer for his comments and suggestions in which, through his appreciation, he demonstrates that he has understood the efforts made to improve a work in which, in addition to the number of samples examined, there is an intense collaboration among EU countries directed towards the management of biodiversity and its conservation.
